# Patient and health-care worker perspectives on the short-course regimen for treatment of drug-resistant tuberculosis in Karakalpakstan, Uzbekistan

**Shona Horter**[1]*, **Jay Achar**[1], **Nell Gray**[1], **Nargiza Parpieva**[2], **Zinaida Tigay**[3], **Jatinder Singh**[4], **Beverley Stringer**[1]

**1** Médecins Sans Frontières, London, United Kingdom, **2** Republican Specialized Scientific Practical Medical Center of Phtiziology & Pulmonology of the MoH of Uzbekistan, Ministry of Health of the Republic of Uzbekistan, Tashkent, Uzbekistan, **3** Republican Phtiziology Hospital #2, Ministry of Health of Karakalpakstan, Nukus, Uzbekistan, **4** Médecins Sans Frontières, Tashkent, Uzbekistan

* Shona_horter@yahoo.co.uk

**Data Availability Statement:** MSF has a managed access system for data sharing that respects MSF's legal and ethical obligations to its patients to collect, manage and protect their data responsibly.

## Abstract

### Introduction

Standard multidrug-resistant tuberculosis (MDR-TB) treatment is lengthy, toxic, and insufficiently effective. New drugs and a shorter treatment regimen (SCR) are now recommended. However, patient and health-care worker (HCW) perspectives regarding the SCR are unknown. We aimed to determine the views and experiences of patients with MDR-TB and HCW regarding the SCR in Karakalpakstan, Uzbekistan.

### Methods

In a qualitative study, we conducted 48 in-depth interviews with 24 people with MDR-TB and 20 HCW, purposively recruited to include those with a range of treatment-taking experiences and employment positions. Data were analysed thematically using Nvivo 12, to identify emergent patterns, concepts, and categories. Principles of grounded theory were drawn upon to generate findings inductively from participants' accounts.

### Results

All patients viewed the SCR favourably. The SCR was seen as enabling an expedited return to work, studies, and "normality". This reduced the burden of treatment and difficulties with treatment fatigue. The SCR appeared to improve mental health, ease difficulties with TB-related stigma, and foster improved adherence. While patients wanted shorter treatment, it was also important that treatment be tolerable and effective. However, HCW doubted the appropriateness and effectiveness of the SCR, which influenced their confidence in prescribing the regimen.

Ethical risks include, but are not limited to, the nature of MSF operations and target populations being such that data collected are often highly sensitive. Data are available on request in accordance with MSF's data sharing policy (available at: http://fieldresearch.msf.org/msf/handle/10144/306501). Requests for access to data should be made to data.sharing@msf.org.uk.

**Funding:** The author(s) received no specific funding for this work, which was carried out within MSF activities.

**Competing interests:** The authors have declared that no competing interests exist.

## Conclusion

The SCR was said to benefit treatment completion and patients' lives. HCW concerns about SCR appropriateness and effectiveness may influence who receives the regimen. These are important considerations for SCR implementation and MDR-TB treatment developments, and dissonance between patient and HCW perspectives must be addressed for successful implementation of shorter regimens in the future.

## Introduction

Multi-drug resistant tuberculosis (MDR-TB), TB resistant to rifampicin and isoniazid, is a global public-health crisis [1]. The standard MDR-TB treatment is lengthy (20–24 months), toxic, and expensive, with resulting challenges for adherence and low cure rates [2, 3]. The MDR-TB treatment landscape is changing, with new drugs and repurposed compounds included in the World Health Organisation (WHO) 2019 TB treatment guidelines offering the potential for more effective, less toxic treatment [3–7].

In 2000, based on limited evidence, WHO recommended lengthy treatment for MDR-TB [8]. It has been argued that the duration of MDR-TB treatment is a major barrier to treatment adherence and programme scale-up [9]. The 2019 guidelines maintain a conditional recommendation that new drugs be administered for 20–24 months, while awaiting findings from randomised controlled trials and operational research on the effectiveness of shorter treatment with new drugs [3]. Since 2016, WHO has recommended a short-course regimen (SCR) of 9–11 months for some people with MDR-TB [10]. Observational studies have reported good outcomes with the SCR [11–14] and the STREAM trial found the SCR non-inferior to the standard of care, with no significant difference in the proportion of severe adverse events but with more frequent unfavourable bacteriological outcomes, including treatment failure [15]. Until further evidence becomes available, clinicians may face difficulties designing effective regimens and navigating treatment guidelines [4].

The SCR enables expedited return to work and social activities for people with MDR-TB, and increases their likelihood of treatment completion and adherence [7]. Health system benefits include reduced treatment costs [16]. However, in areas with high prevalence of resistance to drugs included in the SCR, concerns have been raised about the appropriateness of the regimen [2, 17]. The potential risk of amplification of drug resistance has also been raised [16].

There is a dearth of evidence regarding the preferences and priorities of individuals affected by MDR-TB. This study aims to examine the views and experiences of people with MDR-TB and health-care workers (HCW) regarding the SCR in Karakalpakstan, Uzbekistan.

## Methods

In a qualitative study, we conducted in-depth interviews with people with MDR-TB enrolled to the SCR and HCW involved in its implementation, in Karakalpakstan, Uzbekistan in May-July 2019.

### Study context

In the Republic of Karakalpakstan in northwest Uzbekistan, which has a population of 1.7 million, Médecins Sans Frontieres (MSF) and the Ministry of Health (MoH) have provided TB treatment and care collaboratively since 1998. In 2013, an observational study was conducted

to assess the effectiveness of an SCR for MDR-TB, and the SCR was implemented under routine programme conditions in May 2016.

At the time of this study, the SCR included high-dose isoniazid, pyrazinamide, ethambutol (first line drugs), moxifloxacin, prothionamide, clofazimine, and an injectable agent (capreomycin or kanamycin) (second line drugs). It consisted of an intensive phase of 4–6 months (injectables and high-dose isoniazid) and a continuation phase of 5 months. The treatment regimen offered as standard (as an alternative to the SCR), was comprised of 5–6 drugs for 20–24 months.

## Participant recruitment strategy

Participants were recruited purposively to include a variety of experiences and perspectives relating to the SCR. Patients for inclusion were identified from the project SCR database by SH, following stratifying the sample for age, gender and treatment category. Participants were identified to include a range of treatment-taking experiences, ages, and rural and urban locations (Table 1). Identified patients were then recruited by a gatekeeper from within the project, either a counsellor or a member of the medical team, who introduced the study and asked if the individual would be willing to discuss the possibility of participation.

HCW were identified to include range of clinical roles, locations, and length of involvement with implementing the SCR (Table 2). HCW were approached by a research assistant, who introduced the study and invited participation. Participant recruitment was based on emerging findings, and the number of participants for each group was determined based on evidence of data saturation, i.e. when adding further participants did not generate new insights relating to the topic of exploration. Thereby, theoretical approaches to sampling were adopted in addition to purposive [18, 19].

**Table 1. Patient characteristics.**

| Characteristic | Number of participants |
| --- | --- |
| **All patients** | **24** |
| Women | 12 |
| Men | 12 |
| **Age (years)** | |
| 18–24 | 8 |
| 25–34 | 9 |
| 35–44 | 2 |
| 45–58 | 5 |
| **SCR treatment phase** | |
| SCR under routine programme conditions | 21 |
| SCR under observational study | 3 |
| **Treatment category** | |
| On SCR treatment | 15 |
| Completed/cured | 5 |
| Treatment failure | 3 |
| LTFU | 0* |
| Relapse | 1 |
| Transferred to SoC/on SoC | 4 |

*It was not possible to recruit people who were LTFU; there were six potential SCR LTFU individuals, of whom two were inaccessible (prison; Kazakhstan), three were uncontactable, and one agreed to meet but did not attend the appointment. SCR = short-course regimen. LTFU = lost to follow-up. SoC = standard of care.

**Table 2. Health-care worker (HCW) characteristics.**

| Characteristic | Number of participants |
|---|---|
| **All HCW** | **20** |
| Women | 15 |
| Men | 5 |
| **Role** | |
| Doctor | 11 |
| Nurse | 5 |
| Counsellor | 4 |
| **Employer** | |
| MoH | 12 |
| MSF | 7 |

MoH = Ministry of Health. MSF = Médecins Sans Frontières.

## Data collection and analysis

Ethics approval was received from the Uzbekistan and MSF Ethics Review Boards prior to study commencement. Written informed consent was obtained before interviews.

28 interviews were conducted with 24 SCR patients and 20 interviews with HCW. Interviews were conducted by SH (a trained and experienced qualitative researcher) and a translator trained in qualitative research and experienced in Karakalpak (n = 46), or English (n = 2), according to participant preference. The option of repeat interviews was discussed during the first meeting; individuals were selected for repeat interview to explore emergent themes following initial data analysis. Two patients were interviewed twice and one three times. Interview location is shown in Table 3.

Interviews were based on topic guides, and were conversational, flexible, and participant-led. Interview introduction and topic guides were piloted with initial interviews, and then adapted. The adapted approach amended questions exploring participant preferences for treatment and views about injectables, as participants appeared unaware of the potential harmful side effects (including hearing loss) from injectables, and to provide more information on the study process and the purpose of inviting participation. Interviews with patients explored their experiences of engaging with treatment and care, and treatment preferences. Interviews with HCW explored experiences of diagnosing people with MDR-TB, decision-making for treatment regimens, and experiences of supporting patients. Data collection and analysis were iterative; topic guides were adapted as data collection progressed to test emerging themes.

**Table 3. Interview location.**

| Participant type | Interview location | Number of interviews |
|---|---|---|
| **People with MDR-TB (n = 24) (interviews mean of 60 minutes)** | DOT corner private room | 11 |
| | Outside DOT corner | 10 |
| | Participant home | 3 |
| | Shop | 1 |
| **Health care worker (n = 20) (interviews mean of 80 minutes)** | DOT corner private room | 13 |
| | MSF office | 7 |

*Interview time and location were decided according to participant preference. DOT = directly observed treatment. MSF = Médecins Sans Frontières.

Interviews were audio-recorded, transcribed, and translated verbatim using direct translation or equivalent translation to maintain meaning and integrity. Data were analysed thematically using coding to identify emergent concepts, patterns, and themes. The analytic approach drew on elements of grounded theory, including constantly comparing findings within and between cases and actively seeking discrepancies from majority themes, aiming to generate findings inductively from participants' accounts [18]. Initially, transcripts were analysed manually using open, descriptive, *in vivo* coding, and a coding framework was developed and adapted as analysis progressed using Nvivo 12. Attention was paid to the role of the researcher in shaping the data, and analytical codes and categories were shared with NG and BS to explore potential discrepancies in interpretation.

Results are presented with participant quotes coded by P for a patient-participant or HCW for a health-care worker participant followed by a sequential number. For repeat interviews, a number before the P indicates the interview number.

## Results

Participants perceived the shorter treatment course as having benefits for the mental health of patients while increasing their ability to work, study, and participate in social activities, reducing TB-related stigma, and enabling better adherence to treatment. These benefits are juxtaposed against HCW concerns about the regimen's appropriateness and effectiveness, and their belief that TB is a disease that requires lengthy treatment. Key findings are summarised in Figs 1 and 2.

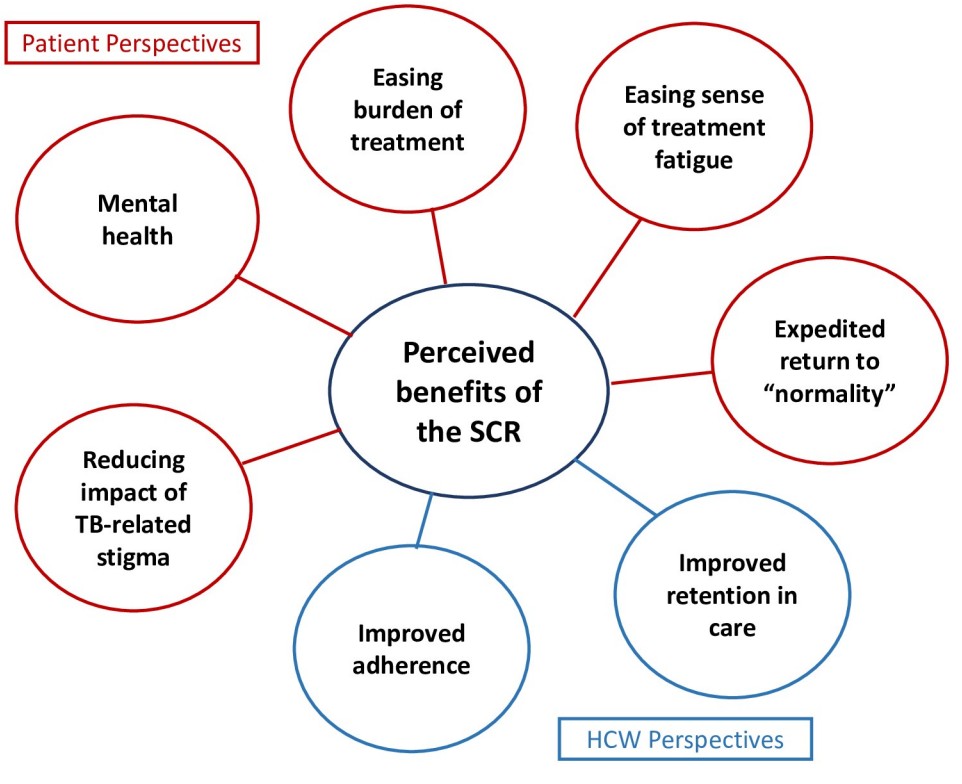

**Fig 1.**

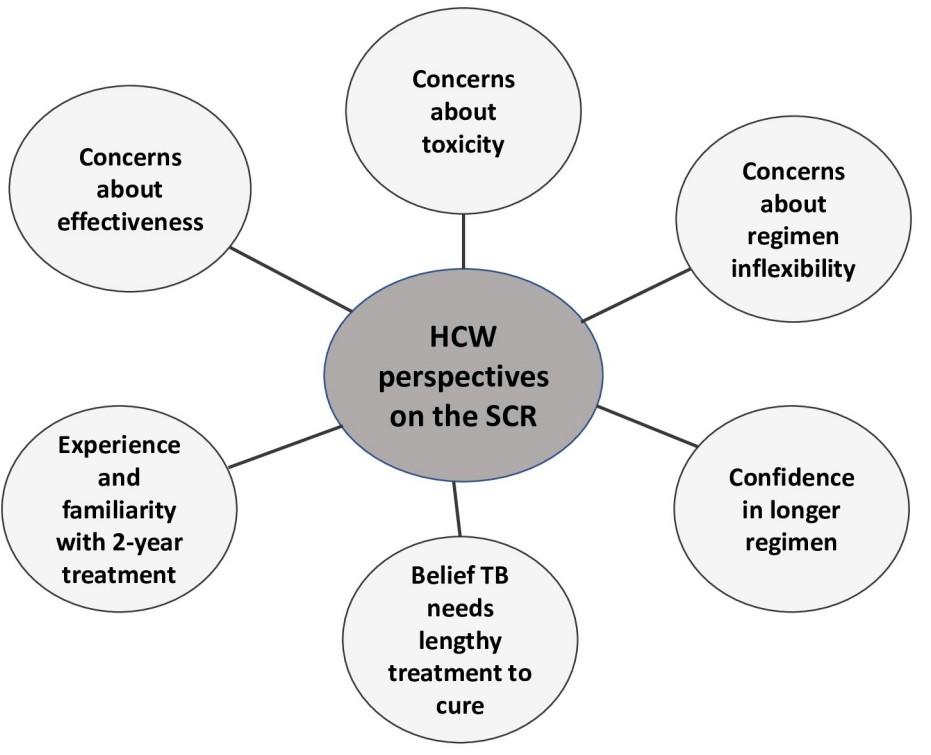

**Fig 2.**

## The benefits of a shorter treatment and ideas for the future

Participants highlighted the ways that a shorter treatment duration influenced the lives of people with MDR-TB. Perceived benefits included the SCR enabling more rapid return to work, studies, and participation in social activities.

> "I would not be able to work for 9 months. Good, OK, I will have a rest for 9 months. All will be good. But, when I was thinking about 2-year treatment, I thought that life will end at that moment. Because I am 25-years-old. According to my list of plans, I would have already been married, and would have a good job, but something went wrong. And, now I am on treatment. I just didn't want to think about 2-year treatment."

01P04

Participants felt the SCR was less likely to cause isolation and discrimination linked to TB.

> "If the duration is 9 months, maybe not all people will know about it, but if the treatment is 2 years then all people will know it. . .There is a discriminative attitude in the family. . . But if the treatment length is 9 months, then it will be easy to socialise at home, to mingle with other members of the family, to talk and eat together. In comparison to when patients are on 2-year treatment when for the whole duration people may feel uncomfortable because of the isolation."

HCW18

Not being able to work, to contribute to family or society, feeling isolated, and navigating the restrictions that TB treatment poses had impacts for mental health and wellbeing and was linked to depression and anxiety. HCW described how this could lead to some considering, attempting, or committing suicide.

"Some of them say "I don't want to live..." he couldn't enter the University; he was at home and thought that he was a burden for his family."

HCW05

The shorter length of treatment was said to ease these impacts, enabling hope, and a sooner return to "normality", which psychologically could be easier for patients to manage.

"Because the treatment length is 9 months, it doesn't create mental problems for patients... because it is a short period of life that he will dedicate to treatment, then the patient goes back to normal life. Thus, it will not be a psychological pressure for the patients."

HCW18

Many HCW participants highlighted the benefit of the SCR in enabling improved adherence and retention in care.

"Adherence was better in 9 months... to be cured in short time was like a big motivation for them. Short length. The drugs seemed easy for them because they had the chance to be cured quickly."

HCW12

"In the SCR they tell themselves that it is a short course, 9 months only, that they can manage it, and they tried. After stopping the injection in the month 4, their motivation will be again elevated. Because when they stop injection, they stop isoniazid also. After 4 months, they start thinking that many of the treatment drugs were reduced. Thus, motivation will increase, and they take treatment well."

HCW15

These benefits are especially important as some patients appeared to develop a treatment fatigue over time, becoming tired of drug side-effects and navigating treatment and wanting a "rest" from treatment.

"I was thinking to have some rest for 1–2 or 3–4 days. I am thinking maybe I will recover after the rest and maybe drug intake will become easier for me... For example, to stop for 1 week to earn money... [because] after taking the drugs there is no energy to work."

P23

"The side-effects... they get fed up with the drugs, and tired of treatment. Because a patient can have 6-month DS [drug sensitive] treatment, after 6 months it becomes 9 months, and it is like non-stop, taking medication, it makes them very nervous."

HCW19

While all patient-participants described a preference for shorter treatment, participants also wanted the treatment to be effective and enable relapse-free cure.

"I want just to be cured, no matter how many days I take them as long as I get cured."

P24

"Interviewer: For example, if 2-year treatment had a higher chance of cure than 9-month, which one would you choose?

P: I would choose 2-year, at that time I didn't select 6-month [clinical trial] because they told me that there was no guarantee."

P08

Patients also said it was important that a treatment regimen was tolerable and had fewer side-effects. A reduced pill burden was particularly important for easing tolerance and treatment continuation, and was a greater priority than removing injectables.

"As much as possible for me it is better to have less side-effects. If these drugs don't cause difficulties, people can take them for their whole life. The main thing is the side-effects. If there were no side-effects, it wouldn't be difficult for anyone. They can be easily taken."

P11

"I have to take 18 tablets and after the 6th tablet I don't want to take any other drugs . . . Instead of 3 tablets [it would be better] to create one drug that combines the 3 drugs. To take it in one go. If you take 3 tablets it sticks in the throat. Maybe to put drugs in capsules. To make it easier to take."

P23

"I think it would be nicer if there were no tablets, and there would be only injections. . . With the injection, they just inject and that's all, but tablets. . . you need to swallow and then suffer."

P02

## TB is a disease which needs a long time to cure: HCW perspectives on the SCR

Many HCW had substantial experience treating MDR-TB, which had for decades required treatment of 20–24 months. This contributed to HCW having greater confidence and assurance in the standard regimen.

"The standard regimen is the most reliable regimen. If patients take it well, there will be almost no relapse."

HCW09

"With the 20-24-month treatment there is almost no relapse. . . we trust the 24-month treatment."

HCW20

As the SCR includes drugs already used in the standard regimen, but for a shorter duration, many HCW had concerns about its effectiveness. The expertise that HCW have developed,

situated in their long experience navigating MDR-TB treatment, led many to believe that TB needs a long period of treatment to cure. The perceived insufficient duration of the SCR was particularly felt for patients with high disease burden, described as having a "large process in the lung(s)".

> "I had a worry. We are used to treating for 20–24 months. . . and immediately it is reduced to 9 months. . . of course, questions arose why? Saving money on drug supply?. . . my concern was that what if during that short period, it may not turn out to be good."
>
> HCW03

> "We were explained that the drugs of SCR are the same drugs that we were using before. . . All of them have been used here for a long period of time, there is no new drug inside the regimen. It is just new combination. . . I had a thought that maybe 9 months will be less than needed. Is it enough?"
>
> HCW07

> "TB is a disease which can be cured in long period of time. And if the process is extensive in the lungs, I think 20-months treatment is better. If the process is not extensive then 9-month SCR is OK."
>
> HCW20

HCW also doubted the effectiveness of the regimen for those with MDR-TB, as the regimen includes three first-line drugs (isoniazid, ethambutol and pyrazinamide).

> "We also had questions like why to give first-line drugs to patients who are resistant to first-line. For example, now we don't give a patient the drug for which patient is resistant, in standard care. Doctors had that question."
>
> HCW16

These concerns about SCR effectiveness influenced HCW views on the appropriateness of the SCR, with the regimen said to be only for new cases with a low disease burden. HCW described difficulties with establishing who is a new case if patients were not always open about their treatment history. Additionally, HCW described people accessing TB care late, due to delays in care-seeking. Concerns about toxicity of the regimen led to several HCW feeling it was appropriate only for young patients without comorbidities.

> "The difficulty of prescribing this treatment is the presence of more adverse events than in other types of treatment. . . there are young patients, 20-25-year-old, we can include them easily, but middle-aged ones and elderly patients are difficult to include into treatment."
>
> HCW13

Unfamiliarity with the SCR may exacerbate HCW concerns about its appropriateness and effectiveness. Several HCW said they wanted to know the outcomes and effect of the SCR. As well as wanting to learn from experiences with the SCR in other countries, it appeared important for HCW to see how their patients responded to the SCR to feel confident that the regimen can work in this setting.

"The patients who completed 9-months treatment, how many of them came with relapse? And in how much time was there relapse? What percentage. . . I think there might be more relapse cases in SCR comparing to standard regimen."

HCW09

"Anyway it contains first-line drugs. I cannot say anything until I see the effect, when my patients will take 9 months of treatment and if they radiologically and clinically improve, then I can tell something, but until that time I cannot say anything. The trust will increase when you see the patient improving."

HCW20

HCW described the SCR as inflexible, and many voiced concerns about initiating the SCR and then potentially having to re-start standard treatment, for instance if a drug needed to be changed. HCW described challenges communicating treatment changes to patients and managing their motivation for treatment continuation.

"It is not possible to change drugs of the 9-month regimen, that's why the patient was transferred to standard regimen. . . sometimes due to side-effects of drugs. . . [but] if you stop one drug from the regimen, it is not considered SCR."

HCW14

"Among the patients who were on SCR there were, but not many, cases of failure. . . at that time there were questions from patients. . . why am I included to 2 year?" Then for us it was inconvenient to answer. . . we had troubles with increasing the adherence of patients who we moved to 2 year from the SCR."

HCW13

Patient participants who were transferred to standard treatment from the SCR did not assert losing motivation for treatment-taking on having to re-start a longer treatment course, but they did describe experiencing stress, fear and a sense of hopelessness linked to their previous treatment not having worked and now having fewer available options. One participant appeared to blame herself for her described intermittent treatment-taking when on the SCR, which she felt led to her failing and which she "regretted", as she now had to take treatment for a much longer period of time:

"I started 2 year treatment and it was difficult for me, actually I was supposed to take 9 months' treatment. . . The first result was good but from the second one they started saying that they have to give me 2 year treatment. . . I was very stressed. . . I think it was due to my not taking drugs, many times I used to throw them or hide. . .. in the other treatment you complete your treatment early, and now 2 years' time is not passing, I am starting to hate it but I have to take it."

P14

Many HCW highlighted how they want the best for their patients, portraying a sense of holding responsibility for their wellbeing and cure, and describing the ways they try to support patients.

"The happiest thing is when a patient is cured of TB, for me is the best of, like, joy."

HCW01

"I don't want them to suffer from side-effects and is also good for me to [give them a] successful outcome. First of all, it's for patients to not suffer, and get rid of infection sooner. . . this is my goal. I am a supporter of preventing TB contacts from re-infection in the household."

HCW02

HCW concerns about the effectiveness and appropriateness of the SCR, coupled with their longer experience of managing the standard regimen and their beliefs about the need for lengthy treatment, may influence their confidence in prescribing the regimen:

"In recent times, even if the patient matches low-risk SCR inclusion criteria related to age, condition, bearing in mind the possibility of future intolerance, we are not including patients into SCR, that is it!. . . Our doctors do not want to prescribe SCR."

HCW13

## Discussion

Our findings highlight the benefits of a shorter treatment regimen for people with MDR-TB, supporting their mental health and reducing TB-related stigma, enabling a quicker return to work, school, and social activities, and supporting adherence and treatment completion. All patients wanted a shorter treatment regimen, which was also tolerable, had a low pill burden, was effective, and enabled relapse-free cure. However, HCW concerns about the effectiveness and appropriateness of the SCR reduced their confidence in prescribing the regimen. As far as we are aware, this is the first study to document the perspectives and experiences of people with MDR-TB and HCW about the shorter treatment regimen.

For people with MDR-TB, treatment can feel worse than the disease [20, 21]. The burdens for individuals deriving from navigating disease and experiencing symptoms [22, 23] are compounded by the burden of treatment [24, 25]. What has been termed "structurally induced noncompliance" [25] can result from the weight of the routine work involved in engaging with treatment and care increasing over time, and an individual's capacity to meet such demands becoming overwhelmed. In our study, treatment fatigue developed over time, which undermined motivation for treatment-taking; thus, the SCR was seen as enabling improved adherence. The shorter duration of treatment appeared to reduce its burden, increasing patients' self-efficacy and ability to prioritise and complete treatment. Other means through which to reduce MDR-TB treatment burden and to support people with MDR-TB to overcome challenges with treatment-taking should be investigated in the future.

The SCR was perceived as improving mental health and easing difficulties with TB stigma. A systematic review of psychosocial issues affecting people with MDR-TB found depression, stigma, and psychologically distressing drug side-effects, as well as financial constraints, were common [26]. Being unable to work, and experiencing social isolation and stigmatisation can undermine the sense of identity of people with MDR-TB [27]; in our study, the quicker return to "normality" that the SCR enabled was important.

The legacy of navigating inadequate treatment for MDR-TB [28] may have influenced HCW's belief in our study that TB requires a long period of time to cure. This belief is reinforced by global TB policy, as noted earlier. In our study, the extensive experience of HCWs with the standard treatment led to some feeling less confident with the newer and less familiar SCR.

HCW had concerns about the appropriateness and effectiveness of the SCR, likely reflective of the high prevalence of first-line drug resistance in the region [1]. Additionally, the SCR contains second-line drugs previously used for standard MDR-TB treatment, but for shorter duration, leading to concerns about insufficient treatment length. This was particularly felt to be an issue for patients with high disease burden. Many HCW had concerns about toxicity of the regimen, and therefore doubted its appropriateness for those with comorbidities and of older age. The inflexibility of the SCR added to these concerns, as the need to change any drugs could lead to re-starting the standard regimen.

HCW described wanting the best for their patients. It is understandable that concerns about the effectiveness and appropriateness of the SCR could reduce their confidence in prescribing the regimen. While patients unanimously emphasised a preference for shorter treatment, their main priority was a treatment most likely to achieve cure, as has been reported in South Africa [29]. The length of treatment was less pertinent if the treatment was more tolerable and with reduced pill burden, echoed by evidence demonstrating increased number of drugs increases the risk of loss to follow-up [30]. These findings reinforce the call for MDR-TB regimens to follow previously identified key principles, such as inclusion of at least one new drug class, a good side-effect profile, and maximum duration of 6 months [9].

The familiarity and confidence that HCW displayed regarding previously used, longer regimens, coupled with concerns about the effectiveness and appropriateness of the SCR, highlight the need for tailored support and information for HCW alongside the implementation of new treatment approaches. This will also likely be relevant as settings move forward with piloting and implementing shorter regimens containing new drugs, and all-oral bedaquiline-containing regimens. Our findings suggest that it can take time for practitioners to build trust and confidence in new regimens, and that theory, practice and mentorship may be important in facilitating this process.

## Limitations

Our findings reflect the perspectives of patients who received the SCR, and may therefore reflect more favourable views than for example those lost to follow up. The majority of patient-participants were currently on treatment, and there were challenges with recruiting people who had taken treatment within the SCR pilot (2013–2015). The findings should therefore be interpreted with this in mind. Interviews were conducted 6 years after the SCR pilot began, and 3 years after it became a standard of care. Perspectives on the SCR may change as experience and evidence relating to shorter regimens become more established over time. The interviewer could have been associated with the health programme, which may have influenced participant accounts, potentially making it more challenging to access layers of narrative beyond those deemed to be socially acceptable. However, HCW openly shared their concerns and criticisms of the SCR.

## Conclusions

Our study finds that the SCR has perceived benefits for people with MDR-TB. All participants wanted a short treatment regimen that is tolerable and effective, with reduced pill burden. HCW have concerns relating to the effectiveness and appropriateness of the SCR. These concerns could reduce HCW confidence in prescribing the SCR. With SCR implementation, it is imperative that TB programmes address dissonance between patient preferences for a shorter, effective, tolerable regimen and HCW concerns about shorter treatment being of insufficient length to cure TB.

## Acknowledgments

We thank the people with MDR-TB and HCW in Karakalpakstan who contributed to this study and shared their time and stories, enabling us to better understand their experiences with MDR-TB and the SCR. Thank you to the MSF project staff who supported implementation of the study, with particular thanks to Alpamis Babiniyazov (MSF) for contribution to study planning, implementation and data collection. We thank Sarah Venis (MSF, UK) for providing editing assistance.

## Author Contributions

**Conceptualization:** Shona Horter, Jay Achar, Nell Gray, Nargiza Parpieva, Zinaida Tigay, Jatinder Singh, Beverley Stringer.

**Data curation:** Shona Horter.

**Formal analysis:** Shona Horter.

**Investigation:** Shona Horter, Jay Achar.

**Methodology:** Shona Horter, Nell Gray, Beverley Stringer.

**Supervision:** Beverley Stringer.

**Validation:** Nell Gray, Beverley Stringer.

**Writing – original draft:** Shona Horter.

**Writing – review & editing:** Shona Horter, Jay Achar, Nell Gray, Nargiza Parpieva, Zinaida Tigay, Jatinder Singh, Beverley Stringer.

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
