## [Decision Letter · Decision Letter 0]

25 Jun 2020

PONE-D-20-13322

Patient and health-care worker perspectives on the short-course regimen for treatment of drug-resistant tuberculosis in Karakalpakstan, Uzbekistan

PLOS ONE

Dear Dr. Horter,

Thank you for submitting your manuscript to PLOS ONE. After careful consideration, we feel that it has merit but does not fully meet PLOS ONE’s publication criteria as it currently stands. Therefore, we invite you to submit a revised version of the manuscript that addresses the points raised during the review process.

We look forward to receiving your revised manuscript.

Kind regards,

Amrita Daftary

Academic Editor

PLOS ONE

Journal Requirements:

2. For qualitative studies, PLOS ONE suggests  consulting the COREQ guidelines: http://intqhc.oxfordjournals.org/content/19/6/349 to ensure that all relevant information is provided (in this case we would appreciate more information about: the number  and training of interviewers; how participants were selected; if a pilot study was tested; how data was coded; if bias issues were considered. Moreover, please ensure that the concluding statements of the Abstract and Discussions are supported by the data shown, as the qualitative nature of the study does not allow to draw conclusions on the program's efficacy.

Reviewers' comments:

Reviewer's Responses to Questions

**Comments to the Author**

1. Is the manuscript technically sound, and do the data support the conclusions?

Reviewer #1: Yes

Reviewer #2: Yes

2. Has the statistical analysis been performed appropriately and rigorously? 

Reviewer #1: N/A

Reviewer #2: Yes

3. Have the authors made all data underlying the findings in their manuscript fully available?

Reviewer #1: Yes

Reviewer #2: Yes

4. Is the manuscript presented in an intelligible fashion and written in standard English?

Reviewer #1: Yes

Reviewer #2: Yes

5. Review Comments to the Author

Reviewer #1: This is such an insightful study - very interesting findings, relevant and clearly written. Love seeing more work on patient perspectives being published and, in particular, how they might contrast/differ from those of HCWs. Just a few minor comments:

- would be nice to see a table or schematic that neatly summarizes the main themes (maybe subdivided by patient vs HCW)

- in your findings, you mentioned how HCW had difficulty communicating the change of regimen from SCR to SoC, and how that could demotivate patients; do you have anything from patients that could substantiate (or dispute) that? specifically, what did the 4 patients who were transferred from SCR to SoC say about their feelings/experience of being switched? It'd be interesting to see if the HCW's concerns on this specific issue were shared by the patients.

- in the discussion & conclusion, it was stated that patients more or less unanimously agree the SCR is a good thing; but in fact, your findings highlights an important caveat: it's good as long as it's effective and doesn't lead to eventual failure or relapse. I think that is an important point that is missed in the conclusion when stating quite matter-of-factly that it benefits patients; the issue of efficacy is important, and HCW worry about it - perhaps this isn't actually in dissonance with patients' perspectives, as they also value efficacy over duration (as they said themselves).

- under limitations, you might consider discussing how: 1) the interviews were done mostly while patients were still on treatment; and 2) interviews were done more or less immediately after implementation of SCR. Think about how that could affect the generalizabilty/transferability of the findings

- related to the above comment, might be worthwhile to draw out /focus on the insights re: implementation of SCR; e.g. importance of educating HCW on the new treatment regimen, efficacy, tolerability/AE profile, etc. A main issue raised by HCW is their skepticism around the regimen's efficacy; this highlights the need for HCW education alongside implementation of new regimens. And this has relevance on newer regimens as well.

- finally, again related to previous comment, would be meaningful to add something relevant to how these findings might be applied when thinking about the even newer regimens, like BPaL regimen & all-oral bedaquiline-based regimens.

Thank you for considering my comments.

Reviewer #2: Exploring patient and health-care worker perspectives on the short-course regimen for treatment of drug-resistant tuberculosis at the example of Karakalpakstan, Uzbekistan is a promising development and has a practical impact for further successful implementation of the new WHO guidelines on the DR-TB treatment across the whole region of Eastern Europe and Central Asia. For future research it would be interesting to study as well the factors wich mitigate risks of non-adherence and treatment fatigue (284) outlined in the discussion section.

My suggestion for the current article is to add the full description of the combination of medicines used in the SCR, as well as description of the 1st-line and 2nd-line medicines referred in the texts, as one of the concerns voiced by HCW related to overlap between the medicines used before and now under SCR. I would also like to highlight, that although obtained data and quotes include examples in favour of injectable medicines (193-194) if to compare the volume of pills to be swallowed with injections, I am also aware (from the field work in the region) of different perspectives shared by the patients, with injectables being referred as highly painful and leading to inflammation (requiring catheters) with dreadful toxic effects, including hearing loss. More detailed look into the devastating effects of injectable drugs might be an area for future research as well. However, please feel free to consider if to some extent current suggestions could boost the discussion section.

6. PLOS authors have the option to publish the peer review history of their article (what does this mean?). If published, this will include your full peer review and any attached files.

Reviewer #1: **Yes: **Stephanie Law

Reviewer #2: No

---

## [Author Response · Author response to Decision Letter 0]

20 Oct 2020

Dear Amrita Daftary,

Thank you for your consideration of our manuscript, titled: Patient and health-care worker perspectives on the short-course regimen for treatment of drug-resistant tuberculosis in Karakalpakstan, Uzbekistan. We are grateful of your decision of minor revisions. 

Please find our point-by-point response to each of the reviewers’ comments below, with our response indicated in italics.

Reviewer #1: This is such an insightful study - very interesting findings, relevant and clearly written. Love seeing more work on patient perspectives being published and, in particular, how they might contrast/differ from those of HCWs. Just a few minor comments:

- would be nice to see a table or schematic that neatly summarizes the main themes (maybe subdivided by patient vs HCW)

Thank you for your enthusiastic engagement with our study. We have added two figures to capture the key findings relating to perceived benefits of the SCR and HCW perspectives on the SCR (Figure 1 and 2), which is now signposted in the Results section (line 135).

- in your findings, you mentioned how HCW had difficulty communicating the change of regimen from SCR to SoC, and how that could demotivate patients; do you have anything from patients that could substantiate (or dispute) that? specifically, what did the 4 patients who were transferred from SCR to SoC say about their feelings/experience of being switched? It'd be interesting to see if the HCW's concerns on this specific issue were shared by the patients.

Thank you for this suggestion. We have added data from patient interviews to reflect patients’ perspectives on how changes in treatment regimen and length could affect them. This was described more in terms of the stressful impact of such a change, rather than it undermining adherence/motivation for treatment-taking specifically (as described by HCW). This addition can be found in the Results section, from line 264 onwards:

Patient participants who were transferred to standard treatment from the SCR did not assert losing motivation for treatment-taking on having to re-start a longer treatment course, but they did describe experiencing stress, fear and a sense of hopelessness on their previous treatment not having worked and now having fewer available options as a result. One participant appeared to blame herself for her described intermittent treatment-taking when on the SCR, which she felt led to her failing and which she “regretted” as she now had to take treatment for a much longer period of time:

“I started 2 year treatment and it was difficult for me, actually I was supposed to take 9 months’ treatment… The first result was good but from the second one they started saying that they have to give me 2 year treatment... I was very stressed… I think it was due to my not taking drugs, many times I used to throw them or hide…. in the other treatment you complete your treatment early, and now 2 years’ time is not passing, I am starting to hate it but I have to take it.” P14

- in the discussion & conclusion, it was stated that patients more or less unanimously agree the SCR is a good thing; but in fact, your findings highlights an important caveat: it's good as long as it's effective and doesn't lead to eventual failure or relapse. I think that is an important point that is missed in the conclusion when stating quite matter-of-factly that it benefits patients; the issue of efficacy is important, and HCW worry about it - perhaps this isn't actually in dissonance with patients' perspectives, as they also value efficacy over duration (as they said themselves).

Thank you for raising this important point. We have specified in the conclusion that the findings highlight that the SCR “can” benefit patients, and have reworded the final sentence describing dissonance in perspectives in order to clarify this (lines 355-357):

With SCR implementation, it is imperative that TB programmes address dissonance between patient preferences for a shorter, effective, tolerable regimen and HCW concerns about shorter treatment being of insufficient length to cure TB.

- under limitations, you might consider discussing how: 1) the interviews were done mostly while patients were still on treatment; and 2) interviews were done more or less immediately after implementation of SCR. Think about how that could affect the generalizabilty/transferability of the findings

Thank you for these suggestions. We have added the following to the limitations (line 342-347):

The majority of patient-participants were currently on treatment, and there were challenges with recruiting people who had taken treatment within the SCR pilot (2013-2015). The findings should therefore be interpreted with this in mind. Interviews were conducted 6 years after the SCR pilot began, and 3 years after it became a standard of care. Perspectives on the SCR may change as experience and evidence relating to shorter regimens become more established over time.

We have also clarified that the SCR was first implemented in 2013 within an observational study, and then implemented under routine conditions in 2016 in the introduction (see lines 85-87 below). Therefore at the time of our study, the SCR had already been in place in Karakalpakstan for 6 years since the pilot/study began and for 3 years as a standard of care option.

In 2013, an observational study was conducted to assess the effectiveness of an SCR for MDR-TB, and the SCR was implemented under routine programme conditions in May 2016.

- related to the above comment, might be worthwhile to draw out /focus on the insights re: implementation of SCR; e.g. importance of educating HCW on the new treatment regimen, efficacy, tolerability/AE profile, etc. A main issue raised by HCW is their skepticism around the regimen's efficacy; this highlights the need for HCW education alongside implementation of new regimens. And this has relevance on newer regimens as well.

- finally, again related to previous comment, would be meaningful to add something relevant to how these findings might be applied when thinking about the even newer regimens, like BPaL regimen & all-oral bedaquiline-based regimens.

Thank you for these useful suggestions, we have added the following to the discussion section lines 333-339:

The familiarity and confidence that HCW displayed regarding previously used, longer regimens, coupled with concerns about the effectiveness and appropriateness of the SCR, highlight the need for tailored support and information for HCW alongside the implementation of new treatment approaches. This will also likely be relevant as settings move forward with piloting and implementing shorter regimens containing new drugs, and all-oral bedaquiline-containing regimens. Our findings suggest that it can take time for practitioners to build trust and confidence in new regimens, and that theory, practice and mentorship may be important in facilitating this process. 

Thank you for considering my comments.

Thank you for your engagement with our study, and for your constructive comments which we hope will strengthen the manuscript. 

Reviewer #2: Exploring patient and health-care worker perspectives on the short-course regimen for treatment of drug-resistant tuberculosis at the example of Karakalpakstan, Uzbekistan is a promising development and has a practical impact for further successful implementation of the new WHO guidelines on the DR-TB treatment across the whole region of Eastern Europe and Central Asia. For future research it would be interesting to study as well the factors wich mitigate risks of non-adherence and treatment fatigue (284) outlined in the discussion section.

My suggestion for the current article is to add the full description of the combination of medicines used in the SCR, as well as description of the 1st-line and 2nd-line medicines referred in the texts, as one of the concerns voiced by HCW related to overlap between the medicines used before and now under SCR. I would also like to highlight, that although obtained data and quotes include examples in favour of injectable medicines (193-194) if to compare the volume of pills to be swallowed with injections, I am also aware (from the field work in the region) of different perspectives shared by the patients, with injectables being referred as highly painful and leading to inflammation (requiring catheters) with dreadful toxic effects, including hearing loss. More detailed look into the devastating effects of injectable drugs might be an area for future research as well. However, please feel free to consider if to some extent current suggestions could boost the discussion section.

Thank you for reviewing our manuscript and for your suggestions. 

Lines 88-92 of the introduction contain information on the combination of medicines used in the SCR. We have specified which drugs refer to first line drugs and which to second line, as follows:

At the time of this study, the SCR included high-dose isoniazid, pyrazinamide, ethambutol (first line drugs), moxifloxacin, prothionamide, clofazimine, and an injectable agent (capreomycin or kanamycin) (second line drugs). It consisted of an intensive phase of 4-6 months (injectables and high-dose isoniazid) and a continuation phase of 5 months. The treatment regimen offered as standard (as an alternative to the SCR), was comprised of 5-6 drugs for 20-24 months.

This clarification has also been added to line 226-227 of the results:

HCW also doubted the effectiveness of the regimen for those with MDR-TB, as the regimen includes three first-line drugs (isoniazid, ethambutol and pyrazinamide).

We have added the suggestion to conduct future research into the factors which may mitigate risks of non-adherence and treatment fatigue, to the discussion section lines 306-308:

Other means through which to reduce MDR-TB treatment burden and to support people with MDR-TB to overcome challenges with treatment-taking should be investigated in the future.

Regarding patient perspectives on injectables, we have not mentioned the need for further research into the devastating effects of injectables for MDR-TB, as globally there is now a shift to oral-only regimens for MDR-TB. However, our findings suggest that patient priorities may not always be aligned with international priorities, as patients described the most challenging aspect of treatment as being the oral pill burden, whereas injectables (as a means through which to administer drugs) were perceived as strong and giving direct access to the body. This highlights the need to also consider future treatment options with reduced pill burden, to support patients with tolerating treatment-taking.

We hope that you will find these changes to be satisfactory, and look forward to hearing your final decision in due course. 

Yours sincerely

Shona Horter, PhD 

On behalf of the authors

---

## [Editor Report · Decision Letter 1]

2 Nov 2020

Patient and health-care worker perspectives on the short-course regimen for treatment of drug-resistant tuberculosis in Karakalpakstan, Uzbekistan

PONE-D-20-13322R1

Dear Dr. Horter,

We’re pleased to inform you that your manuscript has been judged scientifically suitable for publication and will be formally accepted for publication once it meets all outstanding technical requirements.

Kind regards,

Amrita Daftary

Academic Editor

PLOS ONE
---

## [Editor Report · Acceptance letter]

13 Nov 2020

PONE-D-20-13322R1 

Patient and health-care worker perspectives on the short-course regimen for treatment of drug-resistant tuberculosis in Karakalpakstan, Uzbekistan 

Dear Dr. Horter:

I'm pleased to inform you that your manuscript has been deemed suitable for publication in PLOS ONE. Congratulations! Your manuscript is now with our production department. 

Kind regards, 

on behalf of

Amrita Daftary 

Academic Editor

PLOS ONE